# AAO-Assisted Nanoporous Platinum Films for Hydrogen Sensor Application

**Melike Sener** [1], **Orhan Sisman** [2] and **Necmettin Kilinc** [1,*]

1    Department of Physics, Faculty of Science & Arts, Inonu University, Malatya 44280, Türkiye
2    Center for Functional and Surface Functionalized Glass, Alexander Dubcek University of Trencin, 911 50 Trencin, Slovakia
*    Correspondence: necmettin.kilinc@inonu.edu.tr; Tel.: +90-422-377-4195

**Abstract:** The effects of the porosity and the thickness on the ability of hydrogen sensing is demonstrated through a comparison of compact and nanoporous platinum film sensors. The synthesis of anodic aluminum oxide (AAO) nanotubes with an average pore diameter of less than 100 nm served as the template for the fabrication of nanoporous Pt films using an anodization method. This was achieved by applying a voltage of 40 V in 0.4 M of a phosphoric acid solution at 20 °C. To compare the film and nanoporous Pt, layers of approximately 3 nm and 20 nm were coated on both glass substrates and AAO templates using a sputtering technique. FESEM images monitored the formation of nanoporosity by observing the Pt layers covering the upper edges of the AAO nanotubes. Despite their low thickness and the poor long-range order, the EDX and XRD measurements confirmed and uncovered the crystalline properties of the Pt films by comparing the bare and the Pt deposited AAO templates. The nanoporous Pt and Pt thin film sensors were tested in the hydrogen concentration range between 10–50,000 ppm $H_2$ at room temperature, 50 °C, 100 °C and 150 °C. The results reveal that nanoporous Pt performed higher sensitivity than the Pt thin film and the surface scattering phenomenon can express the hydrogen sensing mechanism of the Pt sensors.

**Keywords:** platinum; AAO template; nanoporous; hydrogen sensor; resistive sensor





## 1. Introduction

Nanotechnological materials have garnered a lot of attention due to their superior physical and chemical properties with their high surface area/volume ratio [1]. There are several methods for synthesizing nanomaterials, ranging from top-down to bottom-up approaches, including hydrothermal, solvothermal, thermal oxidation, vapor–liquid–solid (VLS) and many others [2,3]. Despite the versatility of synthesis techniques, it is challenging to tailor the shape of the nanomaterials due to the kinetically driven processes, especially for bottom-up techniques [4]. Template-assisted nanomaterial fabrication is more favorable because it provides higher control and precision over the shape of the nanostructures. One of the well-known templates is anodic aluminum oxide (AAO) nanotubes, which have a high aspect ratio; regular, vertically aligned, stable and controllable pore structure; and offer quick, cheap and simple production [5]. The pore shape, diameter and depth of the AAO nanotubes can be adjusted during anodization by changing parameters such as the electrolyte solution, anodization voltage, current, time, temperature and pH [6,7].

Recent technological advancements have been highly encouraging and optimistic in the shift towards renewable and clean energy sources, replacing fossil fuels. One such promising energy source is hydrogen, which has become a big alternative fuel due to its renewable nature, high efficiency and clean energy production [8]. Hydrogen is not only produced from renewable energy sources, but it also produces energy with zero emissions, making it a clean energy source. Currently, hydrogen is utilized in various applications, such as residential, transportation and industrial, operating under different pressure and

temperature conditions [9]. Hydrogen has the potential to replace traditional fuels in various industries, leading to reduced emissions and a clear environment.

Developing steam reforming, electrochemical and other catalytic hydrogen generation systems, as well as progress in membrane technologies for capturing, storage, and separation processes of hydrogen, have attracted significant attention in the field [10]. Research and development in the field of hydrogen production and storage have increased, leading to new and improved technologies for the efficient production and storage of hydrogen. The trend in the sensor field is now geared towards electroactive metal–organic-frameworks (MOFs) membranes, where hydrogen can be compatible with its smallest molecular size for the aforementioned processes [11,12].

On the other hand, hydrogen is the lightest element ever known and is highly flammable, colorless, odorless and tasteless [13]. To ensure the safe production and distribution of hydrogen worldwide, advanced hydrogen sensors with reliable operation are essential. Thus, it is imperative to develop hydrogen sensors that can detect the presence and concentration of hydrogen below its lowest explosion limit (LEL) of 1% in the air for security purposes [14].

Hydrogen sensors can be designed with various working principles, such as resistive, mechanical, catalytic, electrochemical, optical, acoustic, magnetic and others, depending on the application requirements [15–17]. Among these, resistance-type sensors have been favored due to their ease of use and ability to be integrated into electronic units and portable devices. These sensors are typically fabricated using Pt, Pd, Ni and other metals, and the interaction of hydrogen with these metals causes the adsorption of hydrogen on the surface [18–20]. This is followed by the ionization of adsorbed hydrogen molecules, resulting in the absorption of hydrogen within the tetrahedral or octahedral voids of these transition metals as ionized molecules and dissociated atoms [21]. Because of their high reactivity, these elements are widely used as catalysts for hydrogen gas detection, with Pt being one of the most preferred due to its durability and reactivity [22–24].

In this study, we produced AAO nanotube substrates to enhance the porosity and interaction with hydrogen gas. Different thicknesses (3 nm and 20 nm) of Pt were coated on flat glass and AAO templates to compare the catalytic effects of the surface morphologies on the hydrogen sensing properties of Pt.

## 2. Results and Discussion

### 2.1. AAO Nanopores Formation

Among the conventional techniques for AAO template synthesis, two-step anodization is preferred to achieve better control over nanotube formation. The first step involves electrochemical oxidation for pore formation, and the second step involves an electrochemical etching process for the tube shaping [25]. In this study, however, only the electrochemical etching process with a one-step anodization method was used for the formation of AAO nanopores, as the aim was to create nanoporous (NP) Pt layers to enhance hydrogen detection. The electrochemical etching process was monitored by measuring the current between the Al foil and Pt electrodes. The change in the current during the electrochemical etching process is given in Figure 1a. A possible AAO pore formation mechanism and the chemical reactions on the surface of aluminum foil were described as follows [26].

$$H_2O \rightarrow H^+ + OH^- \tag{1}$$

$$OH^- \rightarrow H^+ + O^{2-} \tag{2}$$

$$Al^{3+} + 3OH^- \rightarrow Al(OH)_3 \tag{3}$$

$$2Al(OH)_3 \rightarrow Al_2O_3 + 3H_2O \tag{4}$$

After the electrochemical etching, the average diameter size of the obtained nanopores was lower than 100 nm. The surface morphology of the synthesized nanopores is shown below in Figure 1b, portraying a remarkable uniformity and consistency of the pore formation.

Notwithstanding the presence of a few negligible local deformations, the overall shape and size were identical and homogeneously distributed throughout the surface, reflecting the efficacy and precision of the etching process.

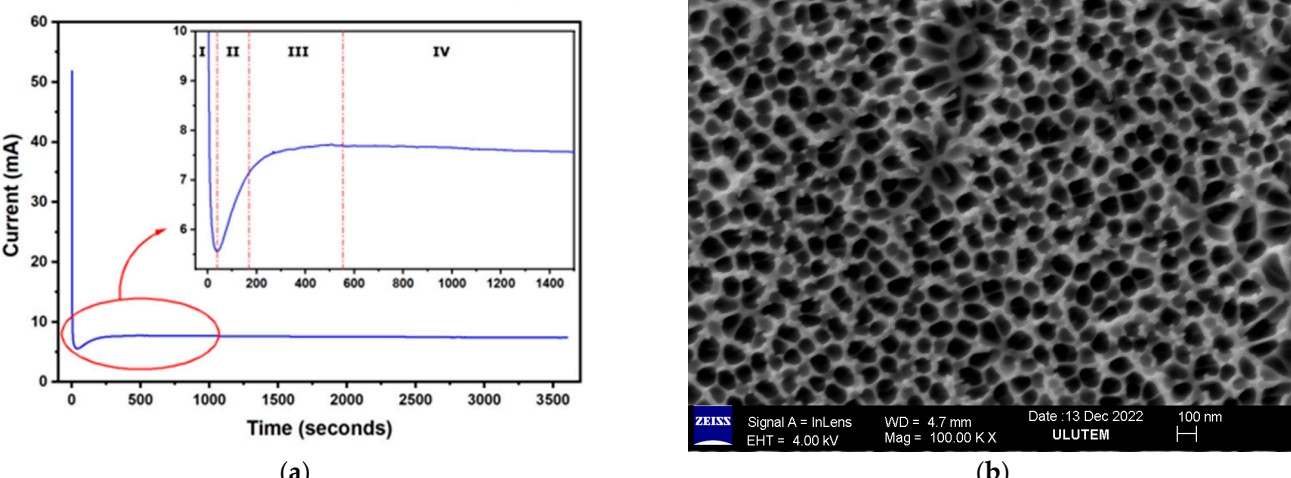

**Figure 1.** (**a**) The current versus time profile during electrochemical etching. (**b**) SEM images of synthesized AAO nanopores.

Jani et al. described in detail the mechanism of AAO template formation [27]. Similarly, the formation of the AAO template was tracked step by step from the current–time curve shown in Figure 1a. Step I represented the initial process of anodization where a constant voltage was applied to deposit a bulk oxide layer on the surface of the Al foil. This caused a drastic reduction in the current value until a turning point. In step II, the applied electric field started dissolving the oxide layer due to the emergence of local electric fields caused by the inhomogeneity and roughness of the oxide surface. In step III, the formation of pores was initiated, followed by their expansion until they completely covered the surface, which can be observed from the linearized curve. In the last step, the pore formation turned into a tube formation with the electric field direction since the oxide formation at the bottom and the oxide dissolution were lasting simultaneously. The linearity of the current–time curve indicated the completion of the AAO template fabrication (Step IV).

*2.2. Pt Thin Films on Glass and AAO Nanopores*

A total of 3 nm and 20 nm Pt thin films were deposited on flat glass and AAO template substrates to compare their hydrogen sensing performances. The continuity of the Pt films was only preserved on the edge of the AAO nanopores (Figure 2), indicating that the structural unity of the NP films directly depended on the nanopore. The surface images revealed that the 20 nm Pt film (Figure 2b) partially covered the surface of the nanotubes.

The EDX measurements revealed the presence of Pt on the AAO nanopore edges in Figure 3 by displaying the quantitative elemental analysis results of bare and 20 nm Pt-coated AAO templates. The inset tables and SEM images in Figure 3a,b show the atomic percentages of the elements on the surface and the images from which the EDX spectrum was taken, respectively. The O, Al and Pt peaks were consistent with the presence of platinum (Pt) and alumina ($Al_2O_3$). The high number of Al and Pt and low number of O counts in Figure 3b are related to the Pt layer on the surface. A comparison of the XRD profiles of bare and 20 nm Pt-coated AAO templates confirmed a similar observation in Figure 3c. The high diffraction peaks at 44.73°, 65.13° and 78.23° (JCPDS#04-0787) matched the Al (200), (220) and (311) crystalline planes, respectively, on both profiles due to the amorphous AAO templates [28]. The characteristic peak at 38.47° belongs to the Al (111) plane and is hindered compared to other planes because the vertical orientation of the nanotubes could be affecting the plane intensities. Similarly, unassigned weak peaks at 40.21° and 42.03° belong to the $\alpha$-$Al_2O_3$ (006) and (113) planes, respectively [29,30]. A

20 nm Pt layer formation on top of the AAO template can be seen by the emerging peak at 39.5° in the focused red pattern.

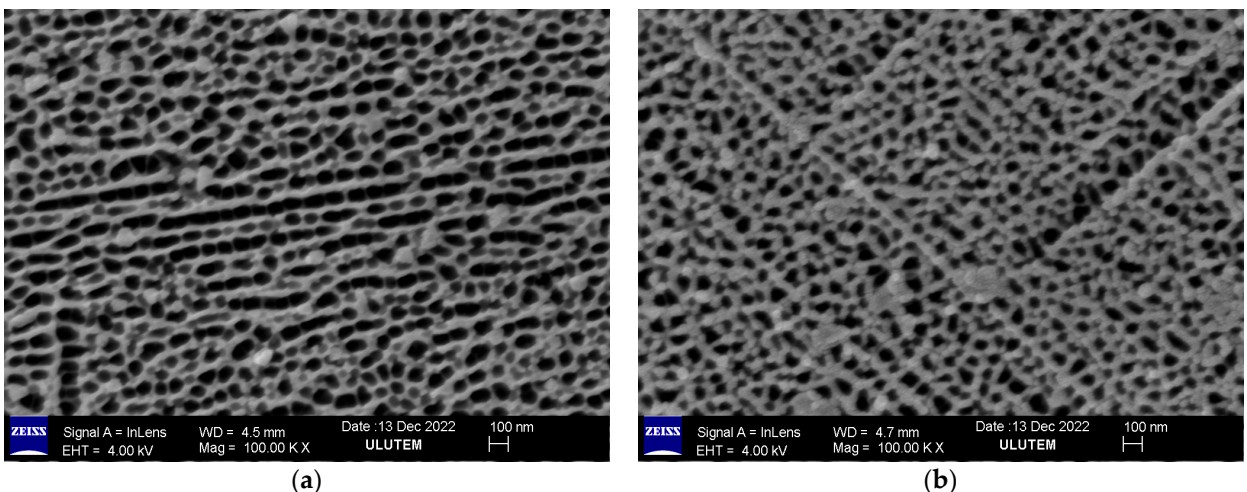

**Figure 2.** SEM images of (**a**) 3 nm and (**b**) 20 nm Pt coated on the AAO substrates.

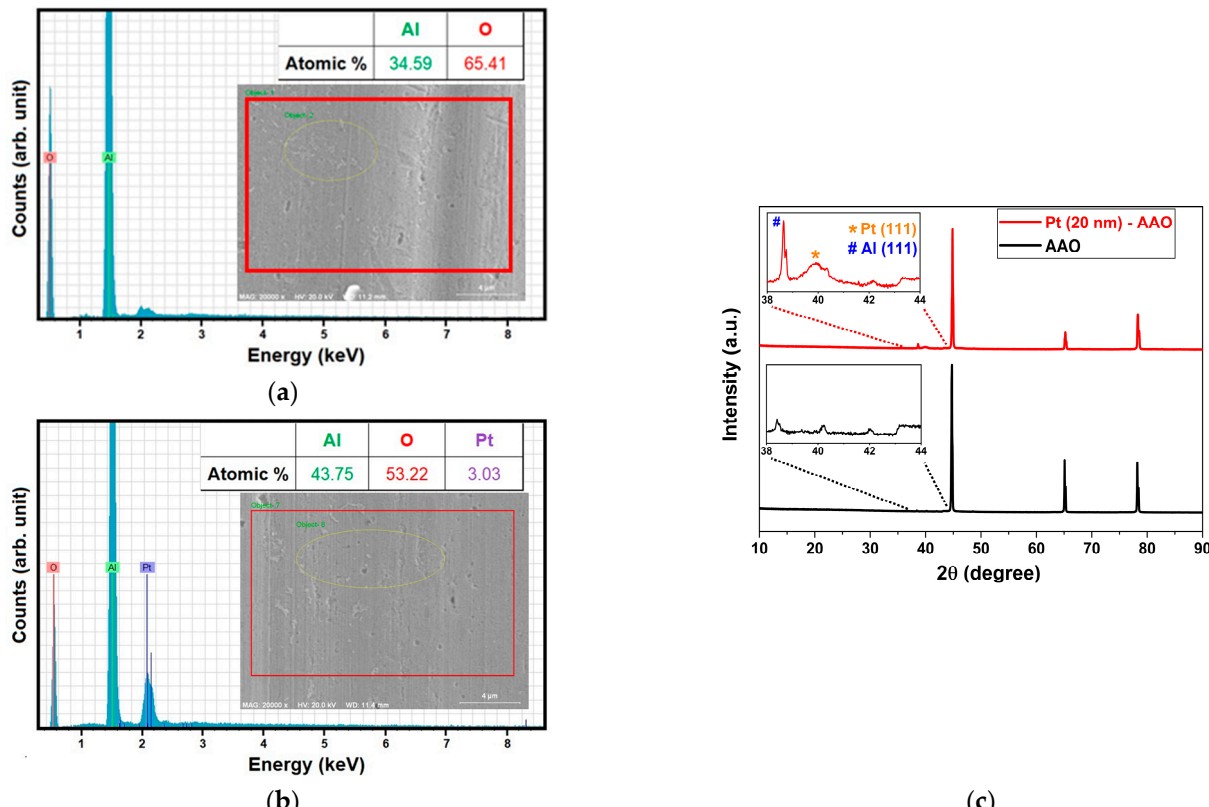

**Figure 3.** EDX elemental analysis of (**a**) bare AAO and (**b**) 20 nm Pt-coated AAO NPs. (**c**) XRD comparisons of the same samples.

### 2.3. Gas Sensing Properties of Pt Films

Isothermal resistance-based hydrogen measurements were carried out at four temperatures: room temperature (RT), 50 °C, 100 °C and 150 °C. The samples were tested for hydrogen concentrations ranging from 10 ppm to 50,000 ppm in a dry air flow. The 20 nm film on the glass substrate showed no response to hydrogen injections at any of the temperatures. Sensor drift was observed during all measurements, specifically at higher temperatures.

Figure 4 compares the dynamic response profiles of Pt TF (3 nm), NP 3 nm and NP 20 nm to 10,000 ppm hydrogen injections at 50 °C and 100 °C. The 3 nm Pt TF has the lowest resistance values. Besides the integrity of the film causing lower resistance compared to the NP layers, the low thickness allows it to be responsive to hydrogen injections by stimulating bulk conduction. A similar relation between the resistance values could be valid for the 3 and 20 nm Pt NP layers. Despite having the highest resistance values, the 3 nm Pt NP sensor exhibited the best hydrogen sensing performance compared to the others.

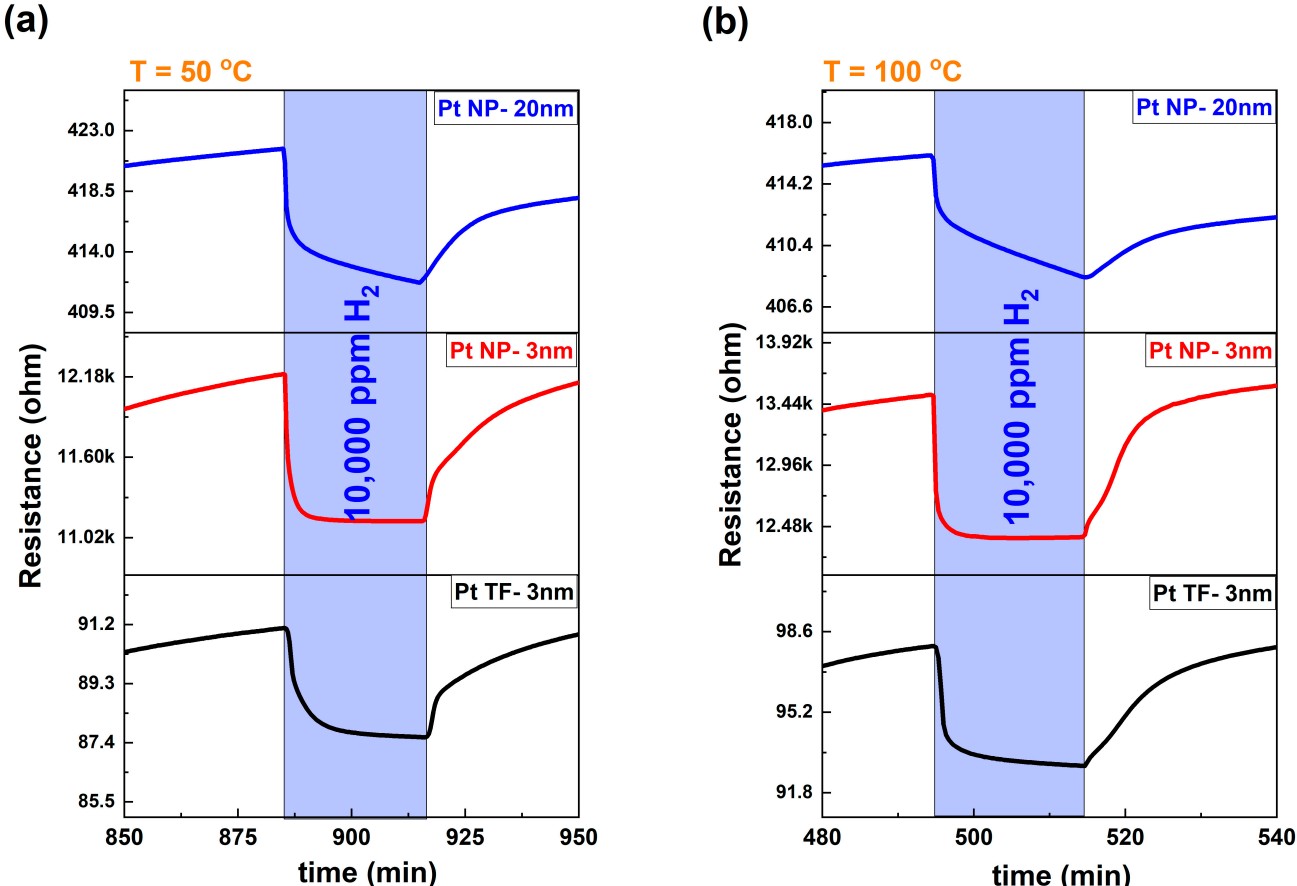

**Figure 4.** Comparison of isothermal dynamic resistance profiles of Pt sensors toward 10,000 ppm hydrogen at (**a**) 50 °C and (**b**) 100 °C.

The hydrogen sensing mechanism of the Pt TF and NP layers was expressed by the surface scattering phenomena. During hydrogen exposure, the number of scattered electrons from the Pt surface was reduced, leading to a decrease in resistance [31–33]. In our study, the catalytic activity of Pt was boosted by the increased surface area/volume ratio from the AAO template, and the low thickness facilitated the surface scattering and, consequently, hydrogen sensing.

To evaluate the sensing capabilities of the sensors, the response values are compared at 50 °C for all the concentrations (10–50,000 ppm) in Figure 5a,b. Prior to performing the response calculations, a baseline substruction process was executed on the dynamic resistance values to ensure clear comparison without any electronic drift issues. In the logarithmic scale, the response vs. concentration relation was found to be linear, except for the Pt TF sensor. It was observed that there was no significant response for hydrogen injections below 1000 ppm, as depicted in Figure 5b. The limit of detection (LOD) values for each sensor were calculated to be approximately 1500 ppm for the Pt TF sensor, 15 ppm for the 3 nm NP Pt sensor and 30 ppm for 20 nm NP Pt sensor, as determined from the cross-section points with the dashed line. This determination was made by accepting the minimum significant sensor response value as 0.5%.

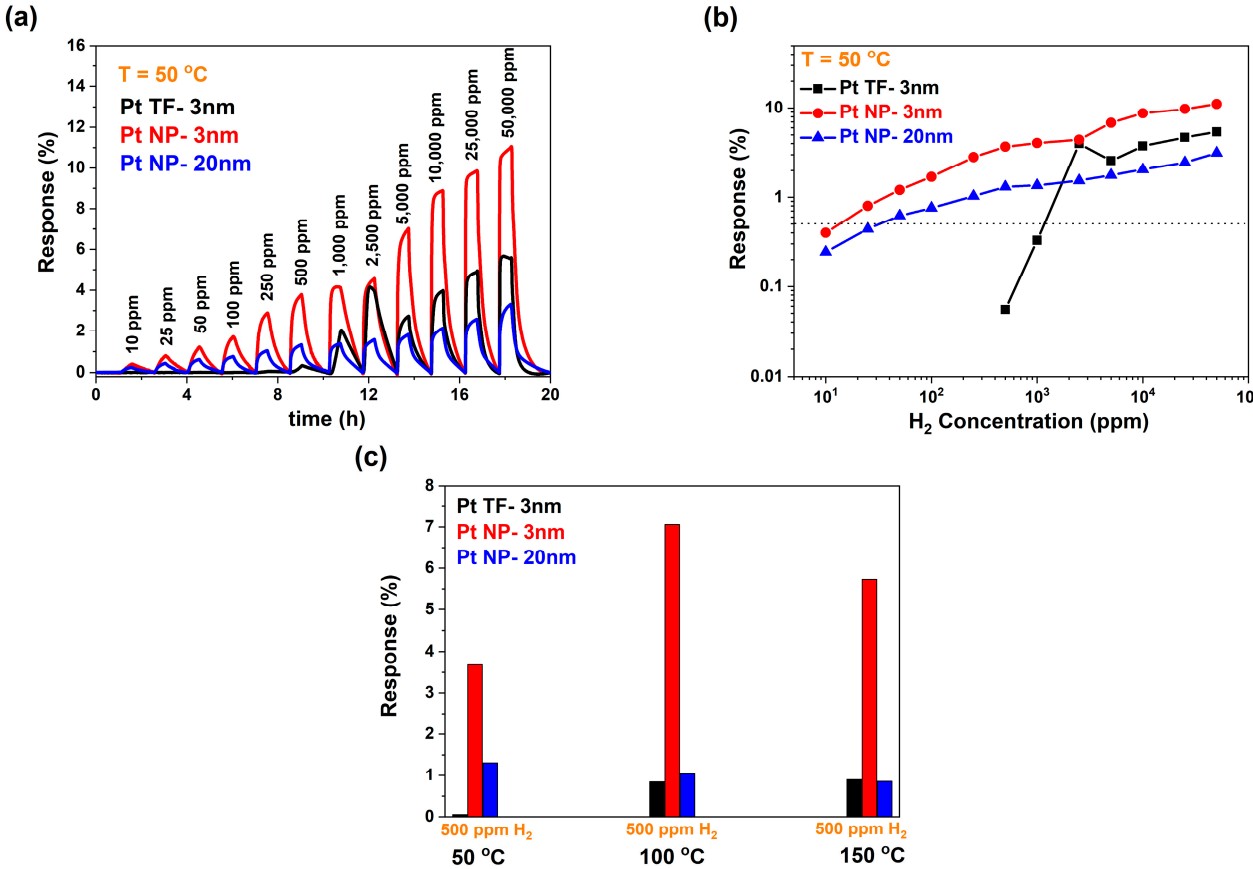

**Figure 5.** Response value comparisons of the Pt sensors for all concentrations at 50 °C vs. (**a**) time and (**b**) log scale concentrations. (**c**) Response value comparison of the Pt sensors only towards 500 ppm hydrogen at 50 °C, 100 °C and 150 °C.

The comparison graph of hydrogen at 500 ppm under varying temperatures (Figure 5c) shed light on the thermal influence on hydrogen sensing in two types of Pt films: the thin film and the nanoporous. The Pt TF showed an improvement in hydrogen sensing when temperatures were elevated, as seen by the increase in response values recorded up to 100 °C, with the exception of hydrogen concentrations exceeding 1000 ppm. This can be observed in Figure 4, where the response values remain consistent even at a concentration of 10,000 ppm. Conversely, the NP Pt films experienced a decline in response values at 150 °C for all concentrations of hydrogen. The probable cause for this decrease may be the diminishing of surface scattering in the NP Pt sensors due to high temperatures and hydrogen concentrations. Despite the instabilities at 150 °C, especially for higher concentrations, the 3 nm Pt NP sensor exhibited superior hydrogen sensing performance, having the lowest LOD and the highest response values at low working temperatures.

Table 1 summarizes the performances of previously reported resistive-based metallic hydrogen sensors and compares them with the AAO template-assisted NP Pt. Yang et al. conducted a comparative study of the sensing properties of single Pt and Pd nanowires in air and reported that there was no resistance change upon exposure to hydrogen in nitrogen, which led them to adopt the surface scattering mechanism to express the sensing mechanism in air [31]. The reversible oxygen/hydrogen adsorption in the surface of 2 nm and 5 nm Pt thin films pointed to the effect of diffusion kinetics, which could be decisive in sensing performances [32]. In Table 1, the best performance belongs to the Pt nanowire arrays, demonstrating their superior catalytic effects and surface interactions in lower dimensions [33]. This highlights the potential of lower dimensional Pt sensors as promising materials for gas sensing applications. With its low operating temperature and high response value, the 3 nm NP Pt sensor exhibited better performance than Pt

and Pt bimetallic thin films. Abburi et al. reported a similar response value for 1000 ppm hydrogen with Pt NP films prepared through the dealloying method [34]. They deduced the effect of the pore size on hydrogen sensing by determining the Knudsen diffusion coefficient. In our case, the thickness and the pore size of the Pt NP films were different, so the hydrogen sensing mechanism was explained with surface scattering phenomena as also previously reported for de-alloyed Pt NP film [35]. The Pt thin films with different thicknesses [36,37] and alloyed with other metals [20,38] showed lower response values, underlying the importance of surface morphologies one more time for surface interactions and sensing processes. On the other side, the alloying of noble metals for thin films [39] and noble metal–Pt heterostructures [23,40] have performed better due to their synergetic effects. In summary, the table concludes that the hydrogen sensing performance of the 3 nm NP Pt sensor is better than most of the reported Pt-based sensors and the effect of surface morphologies, dimensions and synergetic effects are decisive on Pt surface kinetics.

**Table 1.** Performances of resistive-based metallic hydrogen sensors [20]. Modified and updated from the Journal of Alloys and Compounds, 892, N. Kilinc, S. Sanduvac, M. Erkovan, Platinum-Nickel alloy thin films for low concentration hydrogen sensor application, 162237, Copyright (2022), with permission from Elsevier.

| Materials | Ref. Gas | $H_2$ Conc. (ppm) | T (°C) | Response (%) | Ref. |
|---|---|---|---|---|---|
| Pt nanowire | Dry air | 1000 | 277 | 3.5 | [31] |
| 3.5 nm Pt TF | Dry air | 1000 | RT | 2.8 | [32] |
| Pt nanowire arrays | Dry air | 1000 | RT | 5.0 | [33] |
| Pt NP film (dealloying) | Dry air | 1000 | RT | 3.5 | [34] |
| Pt NP film (dealloying) | Dry air | 10,000 | RT | 6.5 | [35] |
| 3.5 nm Pt TF | 5% $O_2$ in $N_2$ | 500 | 200 | 8.0 | [36] |
| 10 nm Pt TF | Dry air | 10,000 | 60 | 1.5 | [37] |
| PtNi alloy TF | Dry air | 1000 | 150 | 4.0 | [20] |
| PtCo alloy TF | Dry air | 10,000 | 25 | 1.2 | [38] |
| Pt/Pd bimetallic ultra-thin film | Dry air | 10,000 | 150 | 13.5 | [39] |
| Pd@Pt core-shell nanocrystal monolayer | Dry air | 1000 | 150 | 2.4 | [23] |
| Pt@Au core-shell nanoparticle assembly | Dry air | 20,000 | RT | 30.0 | [40] |
| 3 nm Pt NP film | Dry air<br>Dry air | 10,000<br>1000 | RT<br>50 | 13.0<br>4.0 | This Work |

## 3. Materials and Methods

### 3.1. Synthesis of AAO Nanotube Substrates

The commercial aluminum foils, which had a purity level of 99.9%, were meticulously cut into dimensions of (2.5 × 1) cm$^2$ and underwent thorough cleaning procedures to ensure optimal results. The cleaning process consisted of 15 min of ultrasonic treatment using acetone, followed by isopropyl alcohol and finally distilled water, in that sequential order. After the cleaning process, the foils were dried under a nitrogen atmosphere to eliminate any residual moisture.

Anodization was then performed for a duration of 1 h, with the foil serving as the anode. An anodization voltage of 40 V was applied at a temperature of 20 °C in a 0.4 M $H_3PO_4$ electrolyte solution, with a platinum counter electrode. As seen in Figure 6, this process resulted in the formation of an anodized layer on the surface of the foil. The anodization process plays a crucial role in determining the properties of the AAO tubes, such as their pore size and distribution.

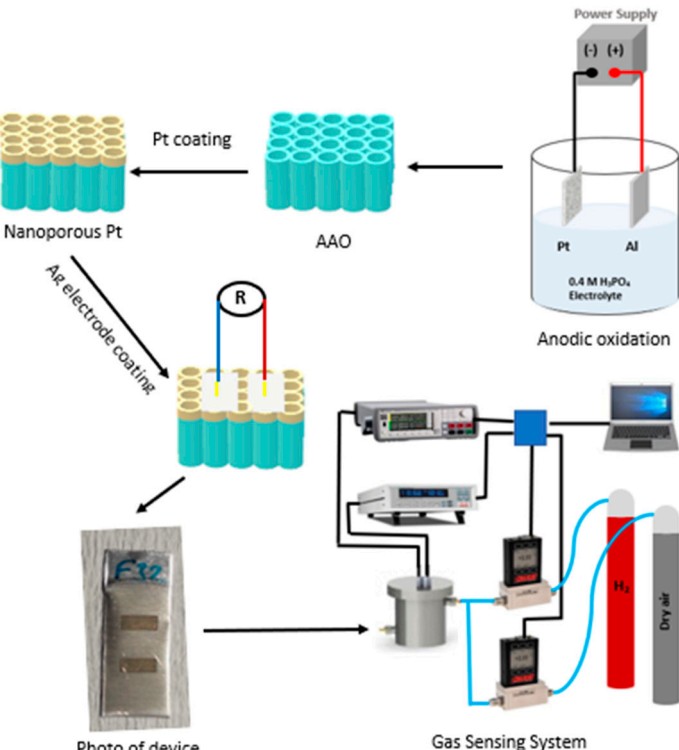

**Figure 6.** A schematic description of the experimental process and setup. The formation of the AAO template by anodization method, Pt coating on AAO with a thin film coating system, deposited Ag electrodes on Pt NP and a photo of the sensor device and the gas measurement system (adapted from [35]).

### 3.2. Synthesis of Pt Films

The flat glass substrates and AAO templates were coated with a thin layer of platinum, with thicknesses of 3 nm and 20 nm, respectively, using the highly advanced thin-film coating system (NVTS 400, NANOVAK, Ankara, Turkey). The precise thickness of the films was meticulously monitored and regulated using a state-of-the-art quartz crystal microbalance (QCM)-based thickness monitor. This ensured that the determined thickness was maintained throughout the coating process.

### 3.3. Characterization of AAO Templates and Deposited Films

The surface morphologies of bare, 3 nm and 20 nm Pt-coated AAO templates were thoroughly monitored using field emission scanning electron microscope (FESEM) (Gemini SEM 300, ZEISS, Jena, Germany). The FESEM was operated at a potential of 4 kV with a working distance ranging from 4.5 to 4.7 mm, providing $100k\times$ magnification. To ensure conductivity for both FESEM and energy dispersive X-ray (EDX) measurements, a 2 nm thick Pd–Au alloy coating was applied onto the pristine AAOs utilizing a sputter machine (SCD 050, BAL-TEC, Los Angeles, CA, USA). Additionally, the crystalline properties of the bare and 20 nm Pt-coated AAO templates were recorded with an X-ray diffractometer (XRD) (Smartlab Diffractometer, Rigaku, Tokyo, Japan). The diffractograms were generated by exposing the samples to X-rays with a wavelength of 1.54059 Å, sourced from a Cu K$\alpha$ source and recorded in a range of 10° to 90°.

### 3.4. Resistive Gas Sensing Measurements

To accurately measure the resistive hydrogen gas sensing properties of the NP Pt and Pt TF layers, a thorough experimentation was conducted. Silver (Ag) was deposited onto the layers by thermal evaporation, utilizing a shadow mask to form two electrodes. The length of the Ag electrodes on all samples was set to 5 mm, with a width of 2 mm and a

distance of 4 mm between two electrodes. A continuous two-point resistance measurement of the sensor devices was performed using a Keithley 2700 multimeter in correlation with the variation of the atmospheric condition in a homemade measurement cell. The cell, made of aluminum with a volume of 1 L, was designed to facilitate gas flow and featured a sample holder that could be heated. The temperature of the sensor devices was precisely controlled using the Lakeshore 335 temperature controller, ranging from 25 °C to 150 °C. All measurement data was recorded using the LabVIEW program in conjunction with a GPIB data acquisition system connected to a personal computer. Two mass flow controllers (Alicat, Madrid, Spain) were employed to adjust the desired hydrogen concentration from 10 ppm to 50,000 ppm. A photo of the sensor device and a schematic illustration of the resistive-type sensor device measurement setup can be seen in Figure 6. The sensor response values were computed by utilizing the resistance values during air flow ($R_a$) and the resistance values during hydrogen injections ($R_g$) through the mathematical formula outlined in Equation (5). To ensure accuracy, a baseline substruction was carried out to account for sensor drift. The baseline resistance ($R_a$) was derived from the subtracted baseline data and used for the calculation of the response values for all concentrations. The overall experiment was carefully executed to provide reliable and precise results on the hydrogen gas sensing properties of the NP Pt and Pt TF layers.

$$\text{Sensor Response } (\%) = \left( \frac{R_a - R_g}{R_a} \right) \times 100 \tag{5}$$

## 4. Conclusions

The study aimed to investigate the effect of surface area/volume ratio and the thickness of Pt films on hydrogen sensing. In this regard, 3 nm and 20 nm Pt films were deposited on flat glass and anodic aluminum oxide (AAO) templates using a facile electrochemical anodization route. The high resolution FESEM images presented that the integrity of the Pt NP films was sustained through the edges of the AAO nanopores. The mechanism of the AAO template formation was also explored by analyzing the current density vs. time profile during the electrochemical process. The EDX measurements confirmed the essence of Pt in the samples by comparing the bare AAO template and the 20 nm Pt NP samples. This was further supported by the emergence of a Pt (111) peak in the XRD profile of the 20 nm Pt NP sample.

After the deposition of Ag transducers for electrical contacts, the sensors were tested for hydrogen at RT, 50 °C, 100 °C and 150 °C using a wide range of hydrogen concentrations. The results show that the 3 nm NP Pt sensor had the best hydrogen sensing performances, specifically at lower concentrations. This was attributed to the enhanced catalytic effect of Pt due to the increased surface area/volume ratio compared to the compact thin film and the low dimension of the NP Pt layer on the AAO substrates, which decreased the surface scattering of electrons and improved the hydrogen detection.

A literature comparison indicated that the AAO template-assisted fabricated 3 nm Pt NP film sensor performed better in hydrogen detection than most of the Pt based de-alloyed and bi-metallic thin films. The different thermo-responsive hydrogen sensing properties of NP and bulk Pt films were attributed to the impact of thickness and compactness on conduction.

In conclusion, the study demonstrated the importance of surface area/volume ratio and the thickness of Pt films in hydrogen sensing and highlighted the significance of using AAO template-assisted fabrication for improved performance.

**Author Contributions:** Conceptualization, N.K.; methodology, N.K.; validation, O.S. and N.K.; formal analysis, M.S., O.S. and N.K.; investigation, M.S.; resources, N.K.; data curation, O.S. and N.K.; writing—original draft preparation, M.S. and O.S.; writing—review and editing, N.K.; visualization, M.S., O.S. and N.K.; supervision, N.K.; project administration, N.K.; funding acquisition, N.K. All authors have read and agreed to the published version of the manuscript.

**Funding:** This study received partial funding from TUBITAK under project number 121M681, Inonu University under project number FCD-2022-2864, and the European Union's Horizon 2020 research and innovation program Project FunGlass with grant agreement number 739566.

**Data Availability Statement:** Not applicable.

**Acknowledgments:** The authors would like to thank ULUTEM and Serdar Özyurt for the FESEM images.

**Conflicts of Interest:** The authors declare no conflict of interest.

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
