# Peer review of "AAO-Assisted Nanoporous Platinum Films for Hydrogen Sensor Application"

_catalysts, doi:10.3390/catal13030459_

Round 1

Reviewer 1 Report

This work should be considered for publication after addressing and revising the following issues:

1. The hydrogen sensor field is quite vast and the introduction does not provide a good overview of the state of the art, mainly because most of the references belong to works published more than 5 years ago. Therefore, please provide a more comprehensive introduction and update the references. 

2. Several details such as the size of the experimental setup shown in Fig.6 are missing. There are no details about sample preparation for SEM and XRD. Please, add more information to the materials and method section. 

3. In the results and discussion section:

3.1 SEM image shown in Fig.1 does not look good. It is a blur. Would it be possible to provide a better-resolution image? The same for Fig.2.

3.2 In Fig. 3, it should be indicated in the SEM image, where the EDX analyses come from. What is the region of the sample? 

3.3. How do you get long-range order for the Pt considering 20 nm nanoparticles? If the material is nanocrystalline, it is impossible to see any coherent reflection by XRD. 

3.4. The hypothesis about the effect of pore sizes on hydrogen sensing via the Knudsen Diffusion coefficient should be better developed. What are the values of such a coefficient and why such a diffusion mechanism would affect the sensing mechanism? 

4. Conclusions and abstract should be re-formulated and should provide a better overview of the work and also the main outcomes. 

Reviewer 2 Report

Self-citations a bit too high - the authors have their own works in total, 23% of all references; request NK to reduce references to his works to 2

Melike Sener      [35]2022

Orhan Sisman     [2]2018

Necmettin Kilinc               [6]2010, [7]2011, [15]2013, [18]2022, [30]2016, [35]2022, [36]2022

NK – 7/39=18%; MS+OS+NK – 9/39=23%

Remarks:

1) Page 1, Line 18 … range between 10 ppm – 5% H2  - Please stick to the same unit of measurement for the parameters used (hydrogen gas concentration) within the article.

2) Page 1, Line 27 VLS - Abbreviation VLS appears for the first time, please decipher.

3) To understand the results described in the publication, a picture of the sensor construction is very much needed.

Reviewer 3 Report

AAO-assisted nanoporous platinum films for hydrogen sensor application

The subject of the paper is using nanoporous aluminum oxide as a template for the nanoporous Pt layer for enhancing hydrogen gas detection utilizing the electron surface scattering phenomena towards Pt surface resistivity.  Through its simple fabrication, the H2 detection achieved a larger response at an optimum thickness of 3 nm Pt layer.  However, some recommendations should be taken into account for publication:

1. Quality of Presentation

The overall presentation quality is acceptable. The article is, on the whole, well-written; the English used in the paper is understandable. However, I have some comments below:

Line 74:

Change "was" to "is".

Figure 3:

The abbreviation "NT" suddenly appeared. It is supposed to be "NP"

Line 142:

Please cite the Figure for the sentence, "There was no significant response lower than 1000 ppm hydrogen injections."

Line 170-172

Replace "rate" with "size", "cm" with "cm2", "respectively" with "sequentially" and "it" with "the foil", which is more suitable.

Equation 5

Write the meaning of variables Ra and Rg

2. Technical review:

"Results and Discussion" section

By comparing the data in Figures 4 and 5, the response at the different temperatures gives similar values (around 7-8%), especially for Pt NP - 3nm in 1% H2. It is better if the author can comment on this behavior not happening in the case of 500 ppm, where the response of one value is at half of the other.

"Materials and Methods" section

In this section, much detailed information should be clearly written such as the electrolyte solution, the Ag electrodes dimension (distance, thickness, area), and the Pt TF dimensions.

Moreover, if the author could provide the optical image of the sample, it will be much better.

3. Overall merit

The article certainly has some merit. For the rest, I believe that the article is organized in a logical and understandable manner.

Round 2

Reviewer 1 Report

All the questions have been answered and the work has been improved. 

I consider that the paper can be published in its present form.